# Hybrid-AI-Based iBeacon Indoor Positioning Cybersecurity: Attacks and Defenses

**DOI:** 10.3390/s23042159

**Published:** 2023-02-14

**Authors:** Chi-Jan Huang, Cheng-Jan Chi, Wei-Tzu Hung

**Affiliations:** 1General Education Center, Ming Chuan University, Taipei 111013, Taiwan; 2R&D Department, THLight Company, Ltd., New Taipei 241407, Taiwan; 3Department of Civil Engineering, National Taipei University of Technology, Taipei 106344, Taiwan

**Keywords:** cyberattacks, BLE, information security, rolling encryption, hybrid-AI-based, iBeacon systems

## Abstract

iBeacon systems have been increasingly established in public areas to assist users in terms of indoor location navigation and positioning. People receive the services through the Bluetooth Low Energy (BLE) installed on their mobile phones. However, the positioning and navigation functions of an iBeacon system may be compromised when faced with cyberattacks issued by hackers. In other words, its security needs to be further considered and enhanced. This study took the iBeacon system of Taipei Main Station, the major transportation hub with daily traffic of at least three hundred thousand passengers, as an example for exploring its potential attacks and further studying the defense technologies, with the assistance of AI techniques and human participation. Our experiments demonstrate that in the early stage of iBeacon system information security planning, information security technology and a rolling coding encryption should be included, representing the best defense methods at present. In addition, we believe that the adoption of rolling coding is the most cost-effective defense. However, if the security of critical infrastructure is involved, the most secure defense method should be adopted, namely a predictable and encrypted rolling coding method.

## 1. Introduction

Beacon is a small data transmitter developed for low-power applications of Bluetooth (4.0 and above). It is suitable for being applied to indoor positioning, which utilizes signal strength to estimate the distance between the Beacon itself and the target mobile phone when locating the user’s position, thus becoming an effective solution for indoor positioning. It solves the problem that the global positioning system (GPS) of a mobile phone is unable to receive satellite signals indoors due to obstruction by the building. More importantly, people consider that Beacon systems are new-generation solutions [1] as they meet the accuracy requirements of indoor positioning at a lower power consumption compared to outdoor positioning systems.

According to the latest research report by Verified Market Research [2], the global market size of Bluetooth Low Energy (BLE) Beacon and iBeacon in 2019 was up to USD 696 million, with an expected growth in significance to USD 109.532 billion by 2027, recording a compound annual growth rate (CAGR) of 87.74%.

To date, various technologies for indoor micro-positioning, including Bluetooth, infrared (IR), ultrasound, radio frequency identification (RFID), Wi-Fi, ZigBee, ultra-wideband (UWB), and others, have been released to commercial markets. Each of them has its own characteristics and features in indoor positioning. Unlike traditional outdoor applications that depend on the GPS, the global navigation satellite system (GNSS), Beidou, and other satellite systems for object positioning, indoor micro-positioning with greater accuracy may collect real-time information via a variety of communication facilities, including satellite systems, various radio frequency systems (cellular, Wi-Fi, Bluetooth), micro-electromechanical systems (MEMS), sensors (accelerometer, gyroscope, and compass), etc. Gradually, applications of iBeacon and BLE Beacon have been increasingly popular on mobile devices, since a majority of these devices have built-in Bluetooth protocol.

On the product end, iBeacon and BLE Beacon have been integrated. For example, Beacon can be found in many camera lenses [3], LED lighting [4], PoS devices [5], and digital billboards [6]; meanwhile, the popularity of smartphones, network devices, and location-based applications [7], as well as the low accuracy of GPS systems in indoor environments, have contributed to the expansion of the global indoor positioning market. The potential applications include locating children [8], the elderly [9], or pets [10] who have been lost; shopping mall navigation [11]; entry, exit, and crowd flow management at large venues (e.g., supermarkets or department stores) [12]; and vehicle management and identification [13]. The higher expansion of iBeacon and BLE Beacon applications often leads to higher risks associated with these applications. At present, the security issues that face the use of Beacon include spoofing, piggybacking and privacy concerns, etc. [14]. Moreover, indoor positioning requests high positioning accuracy and signal availability and accessibility. Chan and Chung in [15] described different attacks against wireless Beacon systems and possible defense mechanisms.

As there is little iBeacon security research involving information security, this study focused on the information security features of iBeacon and introduced possible attacks by hackers and their defense approaches.

The iBeacon system deployed in this study is located in a large area—about 500,000 people pass through Taipei Main Station every day. Therefore, the management unit of Taipei Main Station uses the iBeacon system for business use and provides API to charge externally. iBeacon is public broadcast data, and hackers can easily use mobile phones to collect broadcast information and create fake iBeacon location information. Assuming that it takes 30 min to go around the Taipei Main Station with a mobile phone, it only takes 64 h to build a fake iBeacon system and provide the same service; consequently, the iBeacon built by the institute will be criticized by the public and reviewed by the government. Therefore, the focus of this study is how to design a system that would require a hacker to spend a lot of time and money to crack.

The rest of this paper is organized as follows. Section 2 introduces the background and related work of this study. Section 3 describes the design of the iBeacon system at Taipei Main Station and possible attacks issued by hackers, with defense methods against these attacks. Section 4 concludes this paper and addresses our future studies.

### 1.1. iBeacon-Related Mechanisms

This section introduces iBeacon, the Eddystone protocol, LINE Beacon, and the advanced encryption standard (AES), which are described one by one in the following subsections.

#### 1.1.1. iBeacon

Proposed by Apple, iBeacon is a small transmitter with features of low cost, low power consumption, long component lifetime, and easy deployment. It is applicable to indoor positioning systems. The iBeacon technology enables smartphones or other BLE devices to receive signals/messages from iBeacon systems and execute the corresponding commands. It has features of easy mounting and hardware-free configuration before a mobile phone can join the system and receive the required services.

Basically, any micro-positioning signal/message transmitter using BLE can be identified as a Beacon, and the difference between a Beacon and an iBeacon is the signal frequency they employ [16].

#### 1.1.2. Eddystone Protocol

Eddystone, an open-source system launched by Google, is a communication protocol similar to the Beacon protocol. It also adopts a cross-platform BLE signal format. Eddystone was developed for both iOS and Android operating systems, thus allowing all Bluetooth devices to join this system. Its fields of a data frame for data transmission include a unique identifier (UID), uniform resource locator (URL), telemetry (TLM), and ephemeral identifiers (EID) [14]. All Bluetooth devices can easily follow these fields to communicate with each other.

#### 1.1.3. LINE Beacon

LINE is an instant messaging software. In 2017, its developer launched a Bluetooth transmitter called LINE Beacon, which allows LINE Bot, the chat robot service powered by LINE, to receive information from those nearby devices. Mobile phones with a distance of less than 50 m from a LINE Beacon can receive signals from this LINE Beacon [17].

#### 1.1.4. Advanced Encryption Standard (AES)

In 2001, the National Institute of Standards and Technology formulated AES, the most widely used symmetric key today. Its encryption process is composed of four steps, i.e., SubBytes, ShiftRows, MixColumns, and AddRoundKey [18]. The key lengths of AES can be 128, 192, or 256 bits, depending on the required security level. A key length of 128 bits would require 10 rounds of repeated executions, while that of 192 bits needs 12 rounds, and that of 256 bits would request 14 rounds.

### 1.2. Related Work

#### 1.2.1. Recent Applications of RFID

For enhancing the food management level in India, Biswal et al. [19] presented an RFID-based model to comprehend the key factors affecting the humanitarian supply chain in the warehouse operation of the Indian food storage system. The purpose was to improve warehouse management and inventory efficiency. Jha et al. [20] applied technologies such as RFID and biometric scanning to improve India’s necessities distribution system, addressing the fragility and management flaws of this system and ensuring that necessities could actually be delivered to poor families. Kant et al. [21] took Indian cheese as the research object and proposed a low-cost, non-destructive, and RFID-based method to detect food spoilage.

Focusing on animal safety and conservation in France, Testud et al. [22] designed an automatic device with a detector using RFID to track and record the individual traces of critters in wildlife tunnels to provide valuable suggestions regarding how to maintain these tunnels. To safely protect young salmon passing through the fish-only channel downstream of a river containing a hydroelectric power plant, Tomanova et al. [23] used RFID antennas to detect the behaviors and survival rates of migratory salmon passing via the channel. They confirmed that the two protection systems were significantly effective for young salmon. Bouletreau et al. [24] installed RFID antennas on the turbines of hydropower stations to track and study the predator Silurus glanis and its threat to the endangered species Petromyzon marinus. The results concluded that in the overlapping living environment of the two species, Silurus glanis showed a high rate of predation, potentially endangering the lives of Petromyzon marinus.

Mohammedi et al. [25] proposed a biometrics-based mobile lightweight patient authentication scheme, which was developed by integrating it with information security. Mikhailovskaya et al. [26] introduced the idea of ensuring the stability of information security systems using RFID technology to counter possible attacks through remote scatterers mitigation. To improve the stability of information security in an IoT medical network, Shariq et al. [27] presented an RFID authentication protocol using an encryption system to defend against hacker attacks and avoid personal information leakage.

#### 1.2.2. Safety Promotion of iBeacon

Bai et al. [28] proposed an iBeacon base station containing a Bluetooth 4.0 module and an emergency evacuation system to accurately locate users whose mobile phones were connected to this base station for receiving Bluetooth services, thus enabling them to be guided for evacuation from their current location in case of a disaster. Chen and Liu [29] designed a system to guide the evacuation of people indoors, and track the items left by people during the evacuation through the sensor network composed of iBeacon nodes. Recently, the world’s first indoor navigation and evacuation framework with iBeacon IoT positioning was released to the market [30]. It can calculate the shortest path from the emergency exits to reduce the time for personnel evacuation so as to safely navigate the user groups outside.

Chen et al. [8] proposed a mobile system using IoT devices with components including iBeacon nodes and GPS to locate lost children. From the perspective of prevention, Liu et al. [31] designed a zipper specialized for iBeacon fixation on T-shirts to prevent children from going missing. Lu et al. [32] created a system that guides blind individuals with regard to their spatial location to enhance their safety in the environment they are situated in, and help them have a better quality of life.

#### 1.2.3. Information Security Challenges of iBeacon

Previous studies have examined the integrity and importance of IoT security from different perspectives, including the classification of attacks [33], the timeliness of encryption, algorithm deployment [34], mobile crowdsourcing [35], cloud computing, etc. [36]. Such research on security has also extended their functions by using BLE [37,38]. BLE is a part of Bluetooth 4.0, which is different from the conventional Bluetooth protocols and can be applied to various wearable devices, such as Beacon [39]. However, compared with other BLE technologies, Beacon can be widely applied to various areas/domains due to its low cost and accurate object location identification. Campos-Cruz et al. [40] analyzed the potential threats faced in the practical operation of wireless Beacon systems, and proposed a lightweight cryptography-based security protocol for establishing shared keys. Na et al. [41] revealed the feasibility of attacking iBeacon services via Wi-Fi devices.

## 2. Materials and Methods

Taipei City is the largest political and economic center in Taiwan, and Taipei Main Station, as the transportation hub for foreign travelers entering and leaving Taipei City, is a big rail transportation field including six lines under four different administrations, namely Taiwan Railways Administration (TRA), Taiwan High-Speed Rail (THSR), Airport MRT (mass rapid transit), and three lines of the Taipei Metro, or Taipei MRT. The environments surrounding this field include four underground malls and one highway bus terminal. In 2019, the daily passenger traffic of Taipei Main Station consisted of 86,000 people via high-speed rail, 122,000 via Taiwan Railway, and 319,000 via MRT; the number of passengers taking MRT from various stations was considerably reduced in 2021 due to the impact of COVID-19, yet the passenger traffic at Taipei Main Station remained the highest among all stations at 177,000 passengers [42], totaling 305,000 passengers combined with the daily passengers taking the THSR and TRA. This study therefore adopted Taipei Main Station as the subject in exploring iBeacon security issues.

Due to the complex building structure and the shops and underground malls occupying significant areas of this building, the design of pedestrian passing routes and signs at Taipei Main Station have increased day by day. In order to protect the safety of the public and keep visitors from getting lost, the Taipei City Government launched the app “Taipei Navi” in 2018, which connects to more than 4000 iBeacons deployed across the station to solve issues such as passenger positioning, wayfinding, and commuting. Figure 1 shows the distribution of iBeacons in this station, where 317 and 108 iBeacons are installed on the 1st and 2nd floors, respectively, while 535 iBeacons are set up in the basement.

### 2.1. iBeacon System Design Architecture

The design architecture of the iBeacon system in this study is shown in Figure 2, the number in brackets is the transmission time sequence. First, the Beacon plaintexts (e.g., a restaurant promotion advertisement) are encrypted by invoking the AES algorithm, and then transmitted to mobile phones via the BLE protocol through broadcasting, before the encrypted plaintext is transmitted to the server of the iBeacon system for decryption. After that, mobile phone users may receive the content of plaintext from the server and decide whether to dine at the restaurant or not. Users wishing to have their needs met by one of the promoted restaurants may be guided to this restaurant using the indoor navigation function of the Taipei Navi app.

### 2.2. Positioning Algorithms

The following is our method of positioning mobile phones. The positioning calculation was divided into three stages: initial, estimate, and regression.

#### 2.2.1. Initial Stage

Stage 1, the initial stage, refers to determining the initial point of a mobile phone. The strengths of radio waves received from the mobile phone at the initial point by Beacons 1, 2, and 3 are shown in Figure 3, where W_a = 10^(RSSI_1), W_b = 10^(RSSI_2), and W_c = 10^(RSSI_3) are the signal strengths that Beacons 1, 2, and 3 receive from the mobile phone, respectively. Each Beacon computes its distance to the mobile phone according to the strength of the radio wave; then, the Beacon system employs the triangulation method to determine the position of the mobile phone. The strengths are presented in log scale. The coordinate positioning equations are as follows:(1)$$x_{0}=\frac{\sum_{i=1}^{n}x_{i}W_{i}}{\sum_{i=1}^{n}W_{i}}.\;n=3$$
(2)$$y_{0}=\frac{\sum_{i=1}^{n}y_{i}W_{i}}{\sum_{i=1}^{n}W_{i}}.\;n=3$$

#### 2.2.2. Estimate Stage

Following the completion of the initial point positioning, it was discovered that instant computation of the current location of users was difficult since the Beacon positioning system may only compute the relative position between each Beacon and the mobile phone, while the radio waves required an average based on stable readings. For a better user experience, this system refers to the method of Kok et al. [43] in Stage 2 by acceleration computations run by accelerators. The accelerator, illustrated in Figure 4, was a wireless three-axis accelerator. Currently, hardware devices with built-in acceleration computation only provide acceleration information. In this study, an algorithm was applied to integrate the acceleration from the user’s phone position into a velocity that would receive a second integration to transform the velocity to a displacement, so that a position with the movement direction could be determined according to the gyroscope angle. In addition, Chang et al. [44] pointed out the five main services of the Taipei Main Station intelligent system, which includes the pedestrian indicator guide, i.e., the application of the “Taipei Navi” app, as shown in Figure 5.

#### 2.2.3. Regression Stage

Stage 3 concerns data regression. The results gained from Stage 2, the “estimate” stage, estimated a user’s position once every 30 s. The results were compared with the initial points identified in Stage 1. In addition, the Beacon value prevailed when the error exceeded 15 m, since the errors of the accelerometer were accumulated. For specific Beacons, such as the ones near an escalator, regression was executed immediately upon a numerical strength excess over 70 dB, since the user would soon move to another floor when riding the escalator and they would be in close proximity to Beacons as they came near to the floor. At such a moment, the Beacon radio wave strength would become extremely high, and a rapid regression, also known as quick regression, was required.

### 2.3. Hybrid-AI-Based Positioning Algorithm

In a real field, mobile phones continuously receive values from more than one Beacon. These messages are still unavailable as the strength of signal reception fluctuates due to unstable radio waves, thereby affecting the results obtained via the mathematical positioning algorithm. To solve this problem, this system applied a hybrid-AI-based scheme to our positioning algorithm.

In Stage 1, the hybrid-AI-based positioning algorithm set the location of the mobile phone to an “N × N” square, and the brightness was proportional to the signal strengths received, hence forming a graph.

Figure 6, representing the results of Stage 2, is an example of a four-by-four grid graph. In actual operation, it is a graph composed of 80 × 80 grids, i.e., each positioning position is an 80 × 80 grid. There are 6500 groups of 80 × 80 squares in the railway station. In this study, strength values were stored per second, and the average strengths every seven seconds as one cycle were mapped into this graph, which was immediately submitted to TensorFlow [45]; CNN in TensorFlow was then applied for point identification.

In Stage 3, the hybrid-AI-based positioning algorithm transformed the position of a user’s mobile phone to a radio wave strength graph. The positions of the Beacons with the strongest wave signal were identified as the user’s current position.

## 3. Results

### 3.1. Possible Attacks

In the following section, we describe possible attacks on Beacon systems.

#### 3.1.1. Recording and Use

Figure 7 describes how the code (ID) of the normally working iBeacon system is illegally recorded and stored by hackers. Figure 8 illustrates how a hacker disguises a Beacon to deliver fake messages or broadcast its own services to users. Owing to high hardware deployment and maintenance costs of the Beacon system, the operator will expect the system costs to be partially shared by mobile phone users. At the very least, unauthorized use should be prohibited. However, from a technical viewpoint:Any mobile phone which has a BLE protocol can receive and decode the message from the iBeacon system, meaning any mobile application that can obtain the Beacon broadcasts is able to position the mobile phone. Therefore, hackers may use such a channel to leak the positioning information. That is, a hacker may walk in the field holding a smartphone to fetch field Beacon codes (ID). Any attacker may enter the field open to the public to obtain data without obtaining any permission beforehand.By writing this information into the APP, identification may be mounted to nearby devices without the owner’s agreement. This may leak the user’s location information to the public, making the entire system a target vulnerable to hacker attacks.Company A, investing a huge amount of capital in the deployment of 1000 Beacons at an airport, would like to recover its investment from the location-based service (LBS) provided. However, it is unable to achieve this once the system is utilized by Company A’s competitor. In fact, Company A’s competitor may assign one person to walk into the field holding a smartphone. Then, this person can gain access to this system without prior notice to Company A or without receiving usage agreement from Company A.

#### 3.1.2. Impersonation Attack

As is shown in Figure 9, a Beacon broadcasts messages to the public, and in most cases, these messages are usually not encrypted; therefore, an attacker can detect and replicate the Beacon codes. An attacker can first hack a Beacon and then attack the users connected to this Beacon. He/she may even replicate a faked Beacon in another field using the same ID, when possible. For example, Attacker B sends false messages to users through a faked Beacon. Another example is that Company B uses a Beacon as an attendance assessment device, which would allow a malicious employee to duplicate this Beacon and fake his/her attendance by tricking the Beacon through clock-ins and -outs at abnormal hours, thereby making himself/herself always present at work.

If there is an LBS service, only one faked Beacon needs to be prepared. As is shown in Figure 9, fake Beacons 5 and 4 placed in the same location could cause the app to distribute wrong information in the wrong place. For example, Beacon 5 should send data of place 5; however, a mobile phone will receive the data messages issued by Beacons 4 and 5 at the same time, as Beacon 5 has been wrongly placed in place 4. According to our own experience, one Beacon with maximum power can serve 50,000 passengers at Taipei Main Station in a day. The spread of such misinformation may lead to terrible situations. People receiving place 5 from iBeacon 5 may consider that this place is place 5, instead of place 4. Another example is that iBeacon distributes a message telling passengers that there is currently a disaster nearby, and everyone has to leave immediately, i.e., distributing fake news.

#### 3.1.3. Obfuscation Attack

An obfuscation attack is one of the easiest attacks at the physical level. Hackers put multiple Beacons in the same location. Since applications mostly rely on the Beacon to determine their position and implement behaviors, hackers may use multiple faked Beacons and Beacons under the original system can be placed in the same position, e.g., Beacons 4, 5, and 6 can be placed in the position where Beacon 3 is deployed. As is shown in Figure 10, this may cause serious interference or delay in the transmission of data by Beacon 3 [41]. This, in turn, affects the integrity and correctness of the data, resulting in messages conveying correct information being lost or faked or Beacons triggering wrong behaviors. An impersonation attack does not interfere with the operation of the original system, while an obfuscation attack is a malicious attack with the potential to paralyze the original system.

#### 3.1.4. Recording and Limited Reproduction

The three types of attacks above can be prevented via rolling coding. However, rolling coding can also be attacked by means of limited reproduction. As is shown in Figure 11, when a Beacon has 999 codes, hackers can, at most, record a set of 40 codes, due to time restraints. However, in the case of limited recording, the codes will ultimately be repeated. This type of attack against the Beacon will be found eventually.

### 3.2. Beacon System Defense Methods

This study proposes five methods to defend against the attacks mentioned above.

#### 3.2.1. Data Encryption/Decryption

This method involves simple steps of encrypting Beacon messages and then decrypting them by the receiver; however, such a method is meaningless in the security of the system, since all of the abovementioned attacks may still work as long as the issued messages remain unchanged. The reason for this is that the encryption keys used at different time points would be the same, i.e., the generated ciphertext C would always be the same. Hackers do not need to decrypt C. They can realize that the messages are issued by a specific iBeacon.

For example:Given that fa is the message sent via the Beacon at location A, the ciphertext sent is converted to AES Code AES(fa).(3)
The app receives AES(fa) for further decoding to fa (which is the code indicating location A).(4)
fb → AES(fb), i.e., the app receives AES(fb) and decodes fb (which is the code indicating location B).(5)

The location code of the Beacon at location A, i.e., fa and its ciphertext AES(fa), form a one-to-one relationship. Even hackers cannot solve fa from AES(fa), as the Beacon at location A transmits the same AES(fa) code for fa at all times, hackers can ensure that the current position is place A. Hence, the same AES(fa) continuously sent from the Beacon at point A will ultimately be recognized as place A, meaning that AES encryption will not be defensive.

As is shown in Figure 12, we used the following methods to evaluate the time cost of cracking:a.Impersonation: Assuming that the user only wants to create the illusion of “I am somewhere” (such as for roll call at work, or for the application location service, which is used at Taipei Main Station), a hacker only needs to copy one iBeacon to confuse the system. Assuming that there is no protection, the hacker only takes 1.5 min to copy the position of one iBeacon and carry out an impersonation attack.b.Obfuscation: If a hacker wants to interfere with the data, he/she needs to make 5%, that is, 535 × 5% = 27 fake iBeacon positions, which can interfere with the system operation at a large scale. Assuming that there is no protection, the hacker takes 40.5 min to carry out an obfuscation attack.c.Recording and use: The hacker must record all iBeacon–location relationships before establishing a new system. Assuming that there is no protection, the hacker takes 802.5 min (30 min for one iBeacon) to create a new fake iBeacon system.

#### 3.2.2. Rolling Coding

Rolling coding can be divided into two models, i.e., unpredictable and predictable. The unpredictable model means that the code will not be repeated at all; the predictable code refers to code produced following some rules.

The classic descriptive equation of rolling coding is
f(t) = f(t + 1)(6)

Unpredictable Rolling Coding

In the unpredictable model,
f (t + 1) = g (t, random)(7)

For example, assuming that the Beacon number is expressed by Code t, and “random” refers to a certain random variable that is a random number for enhancing system security, this series of code can be described as
f (t + 1) = g (code, random)(8)

For example, f (t + 1) = (000, 632), and after 10 min, f (t + 1) = (000, 787). The variation from (000, 632) to (000, 787) shows no regularity.

Let g = AES, then
f (t + 1) = AES(code, random)(9)

The random parameter of the encrypted variable code may not be applied to predict the next code result even if AES is cracked.

For example, f (t + 1) = AES (code, random) = YCL2KXHTLDN2XHPA (000, 369)

The decoder may only obtain the code and random variable after decoding AES (code, random). We would like to know the following:Whether the code is shown at a preset reasonable number of digits, e.g., between 1 and 100, which means that the code between 1 and 100 is normal. If the Beacon code exceeds 100, it means that the system has been illegally recorded and is subjected to an impersonation attack or obfuscation attack.Random variable being a random valueAssuming that the hacker has copied three random codes of Beacon location A, if the server S has been hacked and duplicate random in AES (code, random) are found (for example, if the Beacon has 4 codes, all of which are 100), the fake Beacon from the hacker will be found; then, all AES (code, random) is subject to comparison, plus the time stamp T when they are launched, i.e., AES (code, random)∥T, denoted by R, is sent to a mobile phone. On receiving R, this mobile phone sends R to server S to verify whether Τ – Τ′ ≤ ΔΤ, where T′ is the time stamp when S receives R, and ΔT is the longest time for a message to travel from a Beacon to S. The purpose is to confirm whether a replay attack has been triggered.

Hackers or attackers may utilize a recording and use attack, impersonation attack, obfuscation attack, and recording and limited reproduction attack to destroy not only the correctness of LBS broadcasts, but also the relationship between “encrypted messages broadcast” and “the corresponding location”. The detection methods of this type of attack are as follows:(1)Recording and use attack: Following the decoding of false messages which are recorded, it can be discovered that its random value is same as the random value in the original message. For example, hackers record all iBeacon system codes and store them for eventual transmission of false information or to broadcast his/her own advertisement using this iBeacon system. The random values of several messages will be the same, and this type of attack may therefore be detected.(2)Impersonation attack: If an iBeacon issues two messages in which the iBeacon codes are the same, e.g., both are 56, the ciphertexts generated will be different since the random values of the two messages vary. If a recorded ciphertext is broadcast N times, N ≧ 2, it remains unchanged. For example, user C passes by a location and user D, who passes by the same location after 10 min, receives the identical ciphertext AES (code, random) sent by the same iBeacon. This means that what C or D receives is the ciphertext recorded and sent by a hacker. It can therefore be detected that this iBeacon has been subjected to an impersonation attack by a hacker.(3)Obfuscation attack: As per that described for the impersonation attack, N AES (code, random) that S received via the same or different mobile phones are identical, N ≧ 2, indicating that the random values of multiple false Beacons are the same as under an impersonation attack. For example, if N iBeacon-code AES (code, random) received by user E passing by a location is identical to that received by user F, who passed by the same location 10 min later, it shows that this iBeacon has been subjected to an obfuscation attack.The contents of legal messages are constantly changing due to the change in the random variable. In fact, the impersonation and obfuscation attacks mentioned above only record “the same group of messages” or “*N* groups of messages, where *N* ≥ 2”. Therefore, the server would detect that the Beacon is under the abovementioned attack and disable this iBeacon’s false message broadcast.(4)Recording and limited reproduction: Assuming that the user, without knowing the coding mechanism, directly recorded m groups of messages, m < n, where n is the number of iBeacons, if the unpredictable and encrypted rolling coding method was adopted, the data received by the server S of this system could not be applied to identify whether the message was false or legal. However, as the server may detect duplicate random values from the decoded message, S will discover that the Beacon has been subjected to an impersonation or obfuscation attack.

As is shown in Figure 13, in the method of rolling coding, if only one set of data needs to be copied, the effect of copying will be invalid after 15 s. Since the time required to copy an iBeacon is six times the code hopping period of an iBeacon, the cost of cracking is not proportional to the benefit, so the copying will not actually be performed. Of course, a hacker can exhaustively collect the data of each iBeacon, and the time cost is explained as follows:a.Impersonation: Hacker takes 1500 min to copy the location of one iBeacon and carry out an impersonation attack.b.Obfuscation: Hacker takes 40,500 min to crack 27 iBeacons and interfere with the iBeacon system.c.Recording and Use: Hacker takes 802,500 min (30 min for one iBeacon) to crack 535 iBeacons to create a new fake iBeacon system.

2.Predictable and Encrypted Rolling Coding Method

In the predictable model, f(t + 1) in Equation (6) is the code existing in regularity. Generally, the change happens once every 10 min, and the count is incremented by 1, i.e., t + 1. For example, assuming that the coding position is the date, i.e., t, plus 1, and today is the 5th day of a month, t = 5, f(t + 1) = f (6). The purpose is to reduce the risk of the code being accessed and hacked. It also increases the security of the system.

Basically, predictable rolling coding is added with the predictable mark of origin = O(code, index, random), i.e., index. For origin = O(code, index, random) in the original text, the random variable can be omitted. For example, Eddystone uses PDU count/time as an index [46] in Beacon. LINE Beacon not only has a timestamp in the plaintext, but also adopts the SHA256 encryption function to encrypt messages. The key length of SHA256 encryption function is 256 bits. It is hard for hackers to crack this function [47].

The advantage of using the predictable and encrypted rolling coding method is to distinguish whether a message is false or not through timestamps. However, these parameters must be stored in the flash ROM, and the number of flash accesses is limited—excessively frequent accesses will damage the hardware [37]. The system owner must carefully consider the flash’s lifetime and limit the number of changes.

In the meantime, the watchdog/battery replacement may cause the timestamp to be zero in practical management. When detecting an impersonation attack or obfuscation attack, the server must detect whether the constraint Τ′ − Τ ≤ ΔΤ has been met or not. If not, the system administrator shall be warned for further processing. Basically, personnel inspections and active monitoring via an app are probable solutions.

The cracking cost of the predictable and encrypted rolling coding method is the same as rolling coding, and the time cost is explained as follows:a.Impersonation: Hacker takes 1500 min to copy the location of one iBeacon and carry out an impersonation attack.b.Obfuscation: Hacker takes 40,500 min to crack 27 iBeacons and interfere with the iBeacon system.c.Recording and use: Hacker takes 802,500 min (30 min for one iBeacon) to crack 535 iBeacons to create a new fake iBeacon system.

Different from rolling coding, at the moment of impersonation attack, if there is a properly designed code, we can immediately analyze whether the iBeacon is under an impersonation attack.

3.Personnel Inspection

Installing an iBeacon sensor as a part of security personnel’s devices may facilitate the inspection of intrusions or attack behaviors by suspecting individuals patrolling Taipei Main Station and the surroundings. The inspection involves checking whether the time constraint Τ′ − Τ ≤ ΔΤ has been met or not by the server. If not, it means that the Beacon is under impersonation or obfuscation attacks, and the administrative personnel must take charge of excluding the information broadcast from these malicious Beacons.

4.App Active Monitoring

The app Taipei Navi has accumulated more than 100,000 downloads since its release in 2018. The messages received by users using the app from iBeacon are aggregated into the server at the back end of iBeacon. By comparing timestamps, it is possible to see whether an iBeacon is under a replay attack and whether the location of the attacked iBeacon can be identified.

The following experiment assumes that all Beacons can send 1000 signals. The assumptions are as follows:

True positives (TP): How many times attacks are detected as attacks.

True negatives (TN): How many times normal behaviors are detected as such.

False negatives (FN): How many times attacks are detected as normal behaviors.

False positives (FP): How many times normal behaviors are detected as attacks.

(1)Recording and Use ExperimentWe referred to the frameworks shown in Figure 7 and Figure 8 to conduct our recording and use experiment, assuming that the user is using unencrypted Beacon signals, and that the pirated signals are not passing through the user’s node at all. The experimental results are given in Figure 14.

As there was no rolling coding, the server could not detect the problem at all and treated it as a normal signal.

(2)Impersonation Attack ExperimentThe framework shown in Figure 9 is the one we adopted to perform our impersonate attack experiment. Impersonation attack can be detected during inspection. Therefore, the Beacon 5 signal is aborted (however, this also disables the normal Beacon 4). Figure 15 shows the confusion matrix.

Both real and fake signals have been received; there are 1000 normal signals and 1000 fake signals. Without rolling encryption, the 2000 signals have the same password and are treated as attacks at the same time.

(3)Obfuscation Attack ExperimentFigure 10 shows the framework we employed to carry out our obfuscation attack experiment, where Beacon 3 is disabled (however, this will also disable Beacon 6, which is normal). The confusion matrix is given in Figure 16.

Beacons 4, 5, and 6 sending 3000 signals were treated as attacks, while Beacons 4, 5, and 6 with 3000 signals were treated as errors when the original position should have been correct.

(4)Recording and Limited Reproduction ExperimentFigure 11 illustrates the framework we utilized to perform our recording and limited reproduction experiment. Repeated signals were discarded. The confusion matrix is given in Figure 17.

Due to the rolling encryption, the server will receive real and false signals that can be disabled immediately.

5.iBeacon Attack and Defense Comparison Experiment

Table 1 shows the cracking time of the iBeacon attack and defense comparison experiment. The total number of iBeacons in Taipei Station in this study is 535. Assuming that the three attack types are not discovered by any defense, rolling coding, or predictable and encrypted rolling coding method, the times required to crack the different defense methods are different. For example, impersonation takes 1.5 min to crack without any defense. Under the condition that the defense system is not activated, rolling coding and the predictable and encrypted rolling coding method take the same time to crack the attack. However, after the defense system is activated, rolling coding will discover the attack once it suffers an impersonation attack, while the predictable and encrypted rolling coding method can detect the attack immediately without facing an impersonation attack.

## 4. Conclusions and Future Studies

The iBeacon system has been widely used to date and it is available for any mobile device that supports BLE; nevertheless, the question of how to protect its security remains. This study explored the possible attacks on iBeacon launched by hackers and proposed defense strategies. This study did not explore how hackers attack via the MAC address owing to the fact that almost 95% of BLE applications in Taiwan are based on iBeacon and iOS cannot obtain the BLE broadcasting MAC address, making it difficult for hacker attacks to reach their economic scales; in addition, versions after BLE 5 contain various security methods against MAC address attacks [48].

Our experiments demonstrated that, during the early stage of iBeacon system information security planning, information security technology and rolling coding encryption should be included, which represent the best defense methods at present. In addition, rolling coding will discover the attack once it suffers an impersonation attack, while the predictable and encrypted rolling coding method can detect the attack immediately without facing an impersonation attack. In terms of economic considerations, the design cost of rolling coding is low, while the design cost of the predictable and encrypted rolling coding method is relatively high. Therefore, based on the results of this study, we believe that the adoption of rolling coding is the most cost-effective defense. However, if the security of critical infrastructure is involved, such as the Taipei Main Station, the most secure defense method should be adopted, namely the predictable and encrypted rolling coding method.

In the future, we will further study the feasibility of other AI-based technological applications in terms of iBeacon information security. The effect of automatic comparison between the accelerometer and graphs of radio wave strengths received has been described in detail in this work. In the future, these technologies will be integrated into the server to establish an AI-based automatic management platform to be used to defend against hacker attacks.

## Figures and Tables

**Figure 1 sensors-23-02159-f001:**
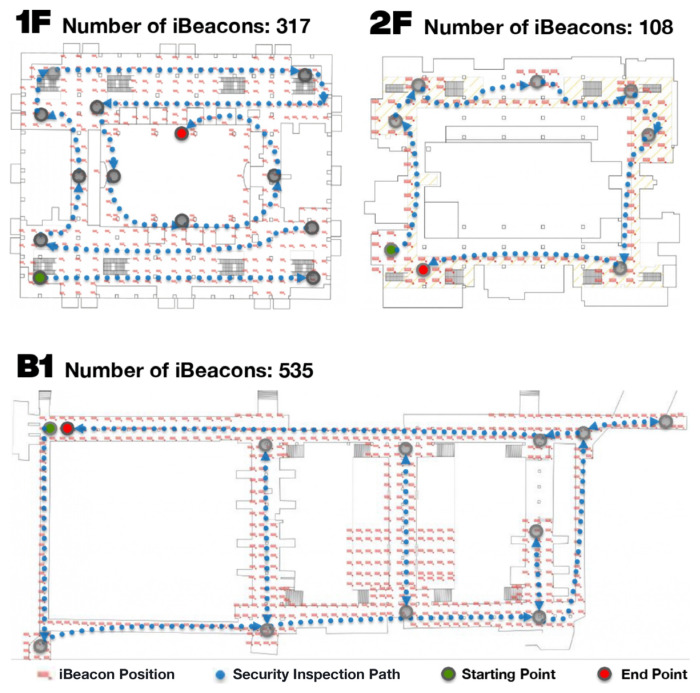
iBeacon locations and regular query route planning at Taipei Main Station.

**Figure 2 sensors-23-02159-f002:**
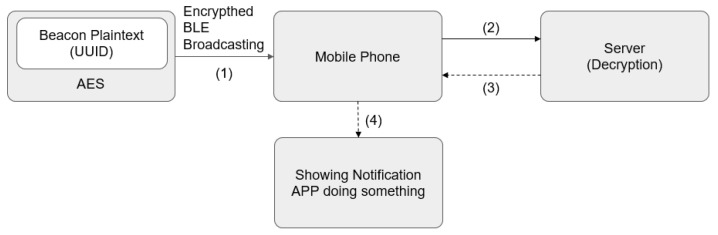
System diagram of the iBeacon system in this study.

**Figure 3 sensors-23-02159-f003:**
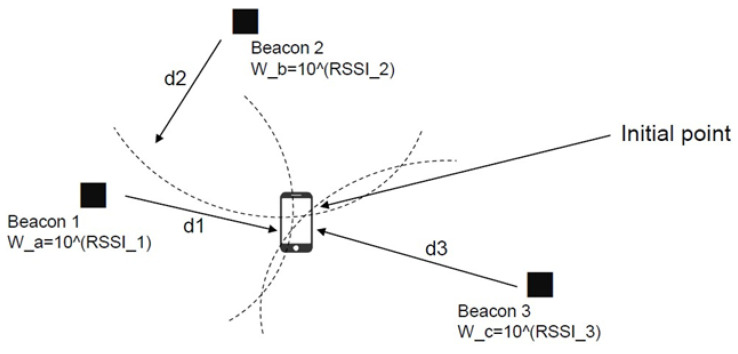
Method of initial point computation.

**Figure 4 sensors-23-02159-f004:**
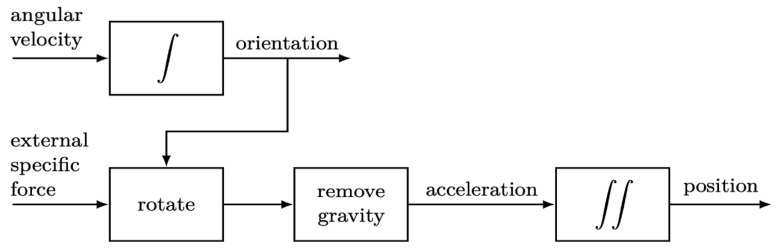
Schematic illustration of dead reckoning, where the accelerometer measurements (external specific force) and the gyroscope measurements (angular velocity) are integrated into the position and orientation of a mobile phone [44].

**Figure 5 sensors-23-02159-f005:**
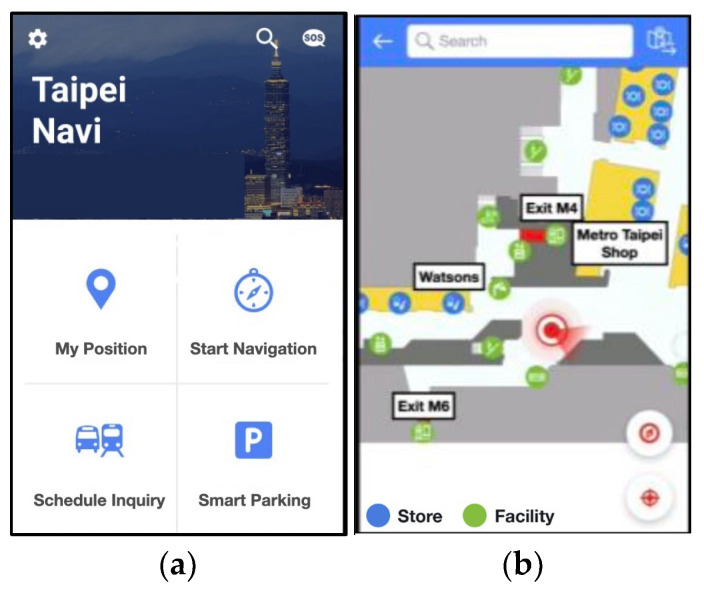
(**a**) Taipei Navi app home screen; (**b**) Taipei Navi app navigation screen.

**Figure 6 sensors-23-02159-f006:**
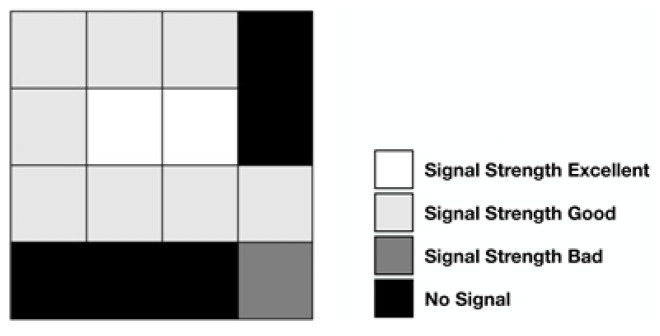
Graph of radio wave strengths received by mobile phones (strengths are indicated in descending order, i.e., white, light gray, dark gray, and black).

**Figure 7 sensors-23-02159-f007:**
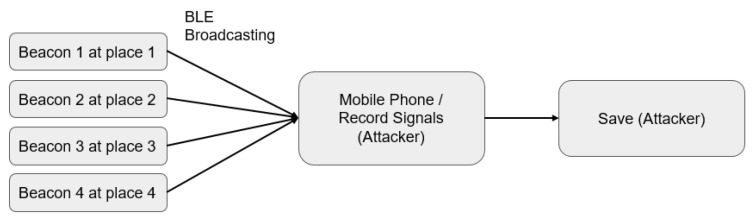
Normal iBeacon codes recorded and stored by hackers.

**Figure 8 sensors-23-02159-f008:**
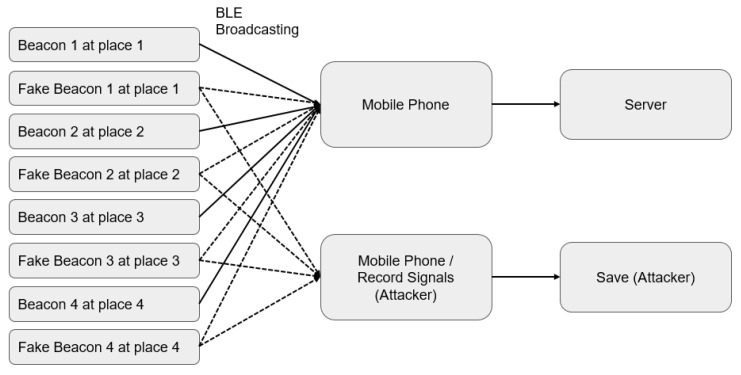
iBeacon hacker sends other users’ messages or broadcasting his/her own services via a mobile phone to disguise as a Beacon.

**Figure 9 sensors-23-02159-f009:**
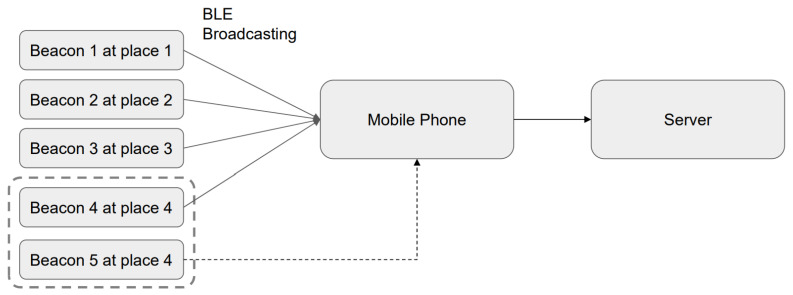
The presentation attack (faked Beacon 5 placed in the place of Beacon 4, leading to delivery of incorrect LBS information).

**Figure 10 sensors-23-02159-f010:**
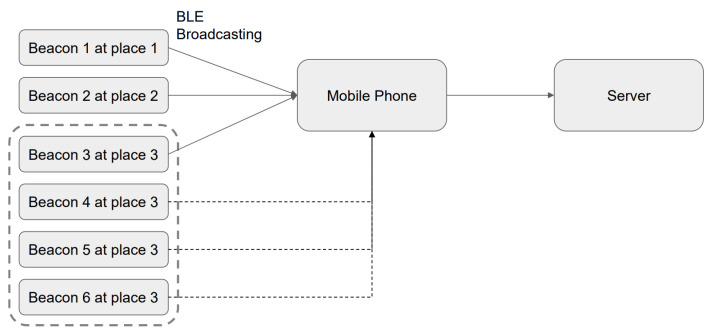
Obfuscation attack (faked Beacons 4, 5, and 6 and Beacon 3 under the original system are all placed at location place 3, sending obfuscation information to users).

**Figure 11 sensors-23-02159-f011:**
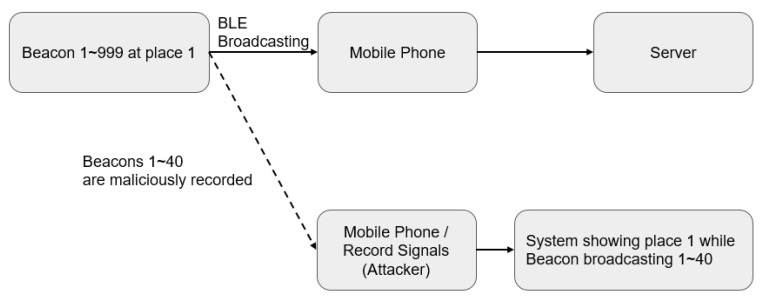
Recording and limited reproduction.

**Figure 12 sensors-23-02159-f012:**
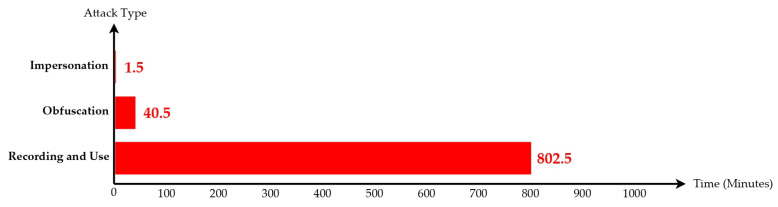
No defense attacks (since AES encryption only has the effect of confusion, it takes only 802.5 min to complete the signal replication and only 40.5 min to start the obfuscation attack; thus, the time required is low).

**Figure 13 sensors-23-02159-f013:**
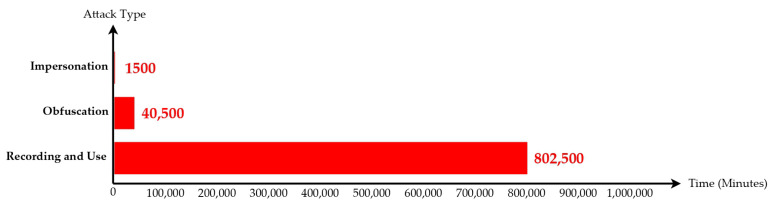
Since recording and use takes 802,500 min to complete signal replication and obfuscation attack takes 40,500 min, the time required is high.

**Figure 14 sensors-23-02159-f014:**
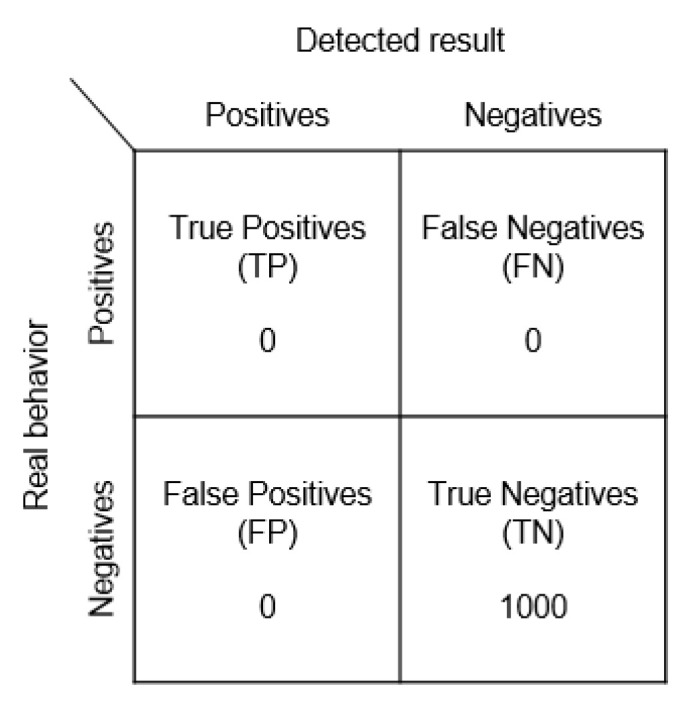
Recording and use confusion matrix.

**Figure 15 sensors-23-02159-f015:**
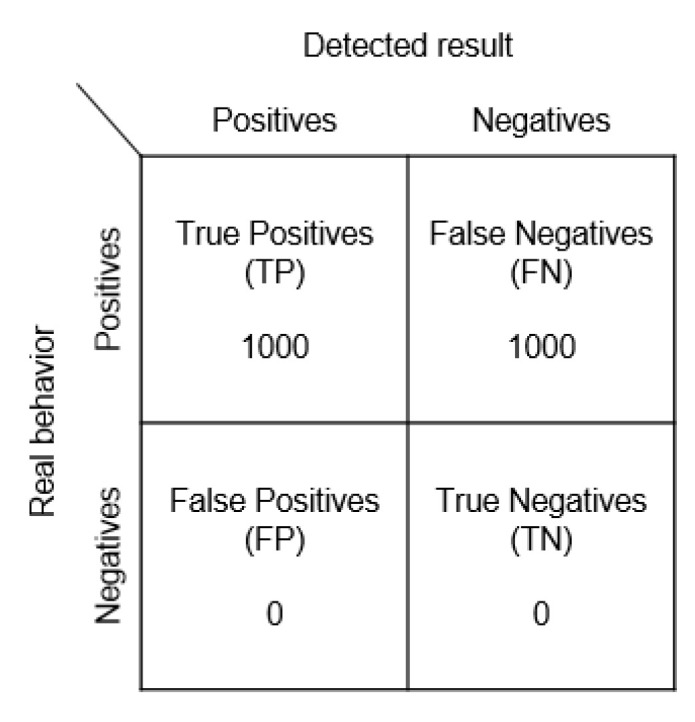
Impersonation attack confusion matrix.

**Figure 16 sensors-23-02159-f016:**
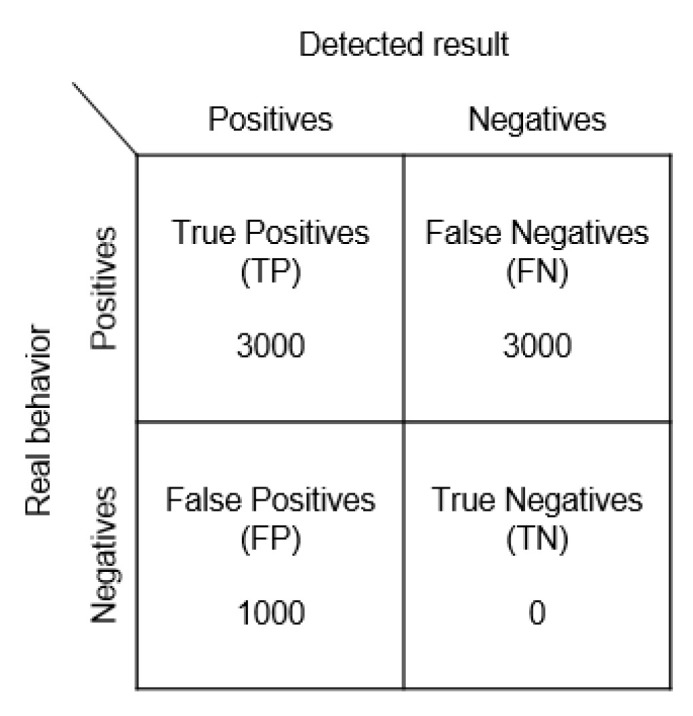
Obfuscation attack confusion matrix.

**Figure 17 sensors-23-02159-f017:**
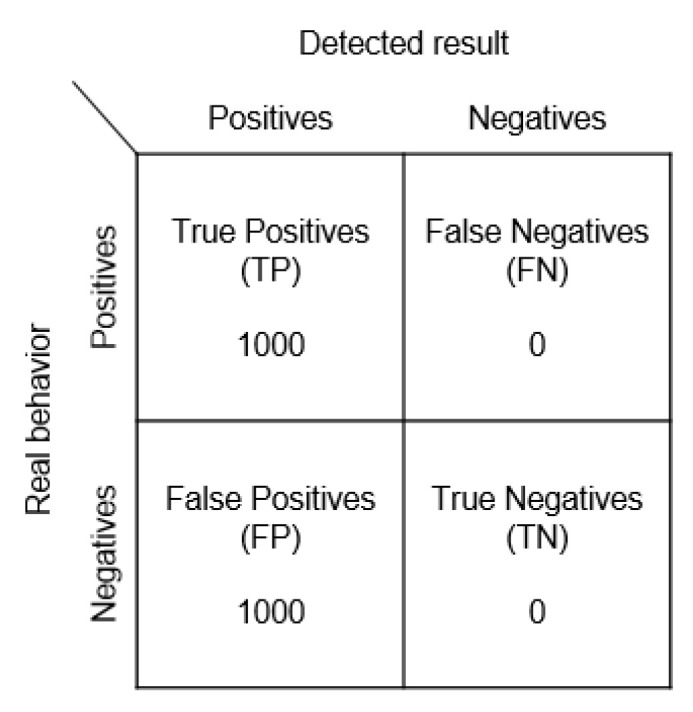
Recording and limited reproduction confusion matrix.

**Table 1 sensors-23-02159-t001:** iBeacon attack and defense comparison.

Attack Type/Defense Method	No Defense	Rolling Coding	Predictable and Encrypted Rolling Coding Method
Impersonation (1 iBeacon)	1.5 min	1500 min	1500 min
Obfuscation (27 iBeacons)	40.5 min	40,500 min	40,500 min
Recording and Use (535 iBeacons)	802.5 min	802,500 min	802,500 min

## Data Availability

The supporting data of this study are available from the author upon reasonable request.

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
