# Peer review of "Hybrid-AI-Based iBeacon Indoor Positioning Cybersecurity: Attacks and Defenses"

_sensors, 2023, doi:10.3390/s23042159_

Round 1

Reviewer 1 Report

The authors present a very interesting research in the AI field to help people go through unknown “roads”, (indoor positioning) namely addressing the Taipei Main station Bluetooth navigation system vulnerabilities to cyberattacks. The paper is very well structured, and the ideas are well explained as well as the overall context to the journal Sensors. The findings of the research is that prior information security planning of a iBeacon system and the rolling coding encryption on its issued messages in Taipei Main Station, are the best defence methods.

Some suggestions to improve the overall quality of the document:

1-     Did the positioning and navigation functions of an iBeacon system have ever been seriously compromised / hacked? Is there information regarding such types of attacks and the consequences? If yes, then the statement in the abstract makes sense (In other words, its security needs to be further considered and enhanced), if not, then the statement should be a suggestion and not a need (”must”)

2-     In the abstract, the conclusion/finding is not very well formulated. The normal reader will not understand what is meant by “the prior information security planning…..”. The rest of the sentence is also too long and not very well self-explanatory. Please try to break that sentence in two, and explain your findings as if it was to a reader that has no idea of what a iBeacon system is or does. What is the prior information? What is the best defence? Are there other defence types? Is the best, however compared to which others? ….. and so on.

3-     The figures 1 and 5 should be improved regarding the quality. Still in Figure 5, two different types of information are displayed. This should be in the legend as a) and b) specified.

4-     In the research the authors do not mention other existing defence systems. Are there any other alternatives in the market available to the one you present in the paper? Did you check and compared the ones that are available with your solution?

The authors write: This study proposes four methods to defense those attacks mentioned above (line 363). I found 5 methods…. Is that correct? Please make clear of how many methods you mentioned. The numbering is also confusing. A good idea here would be to use a table to put the methods with a brief description of each one of them and the major differences (and if possible the limitations and advantages)

3.2.1. Data Encryption/Decryption

3.2.2.1. Unpredictable Rolling Coding

3.2.2.2. Predictable Rolling Coding Method

3.2.2.3. Personnel Inspection

3.2.2.4. App Active Monitoring

5-     In the conclusions is missing the managerial implications and academic implications of the presented research.

6-     Still in the conclusion, is not mentioned the findings of the research as indicated in the abstract …. Regarding the best defence method … Please add this information in a clear way in the conclusions.

7-     The number of references is adjusted to the presented research, as well as the “age” of them (the great majority no older than 5 years). Still, all references look quite adjusted to the journal style-standards. The references are also in ascending order across the document.

Thank you, good job and good luck.

Reviewer 2 Report

The article titled: “Hybrid AI-based iBeacon Indoor Positioning Cybersecurity At-2 tacks and Defenses Thereof”, proposes prior information security planning and rolling coding encryption as the best defense strategy for an iBeacon system. Overall, the article is well-written and readable. Please make the following corrections

Following are the comments/observations that needs to be fixed before publication:

Line 35: Space missing between US$696 and US$109.532. Check for consistency.

Figure 3: has a red underline W_a, W_b, and W_c. Please remove the red underline. I believe it came after printing the image from word.

Line 224-237 and 248-256: Paragraphs not justified.

Figure 6: Add a legend on the right side of the heatmap.

Figure 11: remove the red underline in block “ System showing”

Reference Section: Please add DOI for all the references.

Reviewer 3 Report

In this article the authors are presenting the possible attacks on iBeacon devices issued by hackers, and proposes defence strategies. The present paper is not a scientific article as long as:

-       In the article is not proposed a mathematical model or a theoretical method;

-       no simulations are included in the paper to validate at an intermediate level the hypothesis;

-       measurements in an controlled environment such as an anechoic chamber are not performed;

However, the article presents some interesting aspects regarding the civilian protection. The investigation could increase the interest of the readers if the authors accomplish at list 2 from 3 requests from upside.

Round 2

Reviewer 1 Report

I am satisfied with the updates.

Reviewer 3 Report

As long as the authors did not performed any modification into the article I cannot change any of the recommendations, nor the overall recommendation. I suggest to the authors to highlight very clear into the body of the article their scientific contribution to the mathematical level. It is not clear if the contribution of the authors is presented in the Materials and Methods sections or in the Results section. Also is mandatory to compare their results with the results from other techniques from the literature or with simulations.
